# Self-Supervised High Dynamic Range Imaging with Multi-Exposure Images in Dynamic Scenes

**Zhilu Zhang[1], Haoyu Wang[1], Shuai Liu, Xiaotao Wang, Lei Lei, Wangmeng Zuo[1,*]**
[1] Harbin Institute of Technology, Harbin, China
cszlzhang@outlook.com, cshy2002@gmail.com, wmzuo@hit.edu.cn

## ABSTRACT

Merging multi-exposure images is a common approach for obtaining high dynamic range (HDR) images, with the primary challenge being the avoidance of ghosting artifacts in dynamic scenes. Recent methods have proposed using deep neural networks for deghosting. However, the methods typically rely on sufficient data with HDR ground-truths, which are difficult and costly to collect. In this work, to eliminate the need for labeled data, we propose SelfHDR, a self-supervised HDR reconstruction method that only requires dynamic multi-exposure images during training. Specifically, SelfHDR learns a reconstruction network under the supervision of two complementary components, which can be constructed from multi-exposure images and focus on HDR color as well as structure, respectively. The color component is estimated from aligned multi-exposure images, while the structure one is generated through a structure-focused network that is supervised by the color component and an input reference (*e.g.*, medium-exposure) image. During testing, the learned reconstruction network is directly deployed to predict an HDR image. Experiments on real-world images demonstrate our SelfHDR achieves superior results against the state-of-the-art self-supervised methods, and comparable performance to supervised ones. Codes are available at https://github.com/cszhilu1998/SelfHDR.

## 1 INTRODUCTION

Scenes with wide brightness ranges are often visible to human observers, but capturing them completely with digital or smartphone cameras can be arduous due to the restricted dynamic range of sensors. For instance, during sunset, the sun and sky are substantially brighter than the surrounding landscape, leading the camera sensor to either over-expose the sky or under-expose the landscape. To obtain high dynamic range (HDR) photos in these conditions, exposure bracketing technology becomes a popular option. It captures multiple low dynamic range (LDR) images with varying exposures, which are then merged into an HDR result (Debevec & Malik, 2008; Mertens et al., 2009).

Unfortunately, when the multi-exposure images are misaligned due to camera shake and object movement, ghosting artifacts may exist in the result. Traditional methods to remove the ghosting include rejecting misaligned areas (Zhang & Cham, 2011; Lee et al., 2014; Oh et al., 2014; Yan et al., 2017), aligning input images (TOMASZEWSKA, 2007; Hu et al., 2013; Yan et al., 2019b), and using patch-based composite (Sen et al., 2012; Hu et al., 2013; Ma et al., 2017). With the development of deep learning (He et al., 2016; Dosovitskiy et al., 2020; Liu et al., 2021), recent advances (Kalantari et al., 2017; Wu et al., 2018; Yan et al., 2019a; Liu et al., 2022; Yan et al., 2023a; Tel et al., 2023) proposed training deep neural networks (DNN) for deghosting in a data-driven supervised manner, performing more effectively than traditional ones.

However, DNN-based HDR reconstruction methods usually require sufficient labeled data, each of which should include the input dynamic multi-exposure images and the corresponding HDR ground-truth (GT) image. In order to ensure position alignment between the input reference (*e.g.*, medium-exposure) frame and GT, previous works (Kalantari et al., 2017; Chen et al., 2021; Liu et al., 2023) generally capture the dynamic inputs with the controllable object (generally a person)

---

*Corresponding author.

motion in a static background, and construct GT by merging static multiple-exposure images of the reference scene. Such a collection process is cumbersome and involves high time as well as labor costs, thus limiting the number and diversity of the datasets. To alleviate the need for labeled data, FSHDR (Prabhakar et al., 2021) explores a few-shot manner, and Nazarczuk *et al.* (Nazarczuk et al., 2022) introduce a fully self-supervised approach. The main idea is to construct pseudo-inputs and pseudo-targets for HDR reconstruction. Nevertheless, their performance is unsatisfactory, as the motion and illumination in synthetic LDR images exhibit gaps with real-world ones.

In this work, we aim to reconstruct HDR images directly with real-world multi-exposure images in a self-supervised manner, without synthesizing any pseudo-input data. This objective should be feasible, as most of the information required for HDR results can derive from input data. The property will be more intuitive when HDR color and structure are observed, respectively. Specifically, HDR color knowledge can be inferred from aligned multi-exposure images, and HDR structure can be extracted from some one of the inputs.

We further propose SelfHDR, a self-supervised method for HDR image reconstruction. Inspired by the above data characteristics, SelfHDR decomposes the latent HDR GT into available color and structure components, and then takes them to supervise the learning of the reconstruction network. On the one hand, the color component is estimated from multi-exposure images aligned by optical flow. On the other hand, the structure component is generated by feeding aligned inputs into a structure-focused network, which is learned under the supervision of the color component and an input reference (*e.g.*, medium-exposure) image. Moreover, during the training phase of structure-focused and reconstruction networks, elaborate masks are embedded into loss functions to circumvent harmful information in supervision. During inference, only the reconstruction network is required to predict the HDR result from unseen multi-exposure images.

We evaluate the proposed self-supervised methods using four existing HDR reconstruction networks, respectively. The models are trained on Kalantari *et al.* dataset (Kalantari et al., 2017), and tested on multiple datasets (Kalantari et al., 2017; Sen et al., 2012; Tursun et al., 2016). The results show our SelfHDR obtains 1.58 dB PSNR gain compared to the state-of-the-art self-supervised method that uses the same reconstruction network, and achieves comparable performance to supervised ones, especially in terms of visual effects. Besides, we conduct extensive and comprehensive ablation studies, analyzing the effectiveness of different components and variants.

To sum up, the main contributions of this work include:

- We propose a self-supervised HDR reconstruction method named SelfHDR, which learns an HDR reconstruction network by decomposing latent ground-truth into constructible color and structure component supervisions.
- The color component is estimated from aligned multi-exposure images, while the structure one is generated using a structure-focused network supervised by the color component and an input reference image.
- Experiments show that our SelfHDR outperforms the state-of-the-art self-supervised methods, and achieves comparable performance to supervised ones.

## 2 RELATED WORK

### 2.1 SUPERVISED HDR IMAGING WITH MULTI-EXPOSURE IMAGES

The main challenge of HDR imaging with multi-exposure images is to avoid ghosting artifacts. DNN-based HDR deghosting methods have exhibited a more satisfactory ability than traditional ones. For the first time, Kalanrati *et al.* (Kalantari et al., 2017) collect a real-world dataset and propose a data-driven convolutional neural network (CNN) approach to merge LDR images aligned by optical flow. Wu *et al.* (Wu et al., 2018) utilize the multiple encoders and one decoder architecture to handle image misalignment, discarding the optical flow. Yan *et al.* (Yan et al., 2019a) present a spatial attention mechanism for deghosting. In addition, we recommend Wang *et al.*'s survey (Wang & Yoon, 2021) for more CNN-based HDR reconstruction methods (Prabhakar et al., 2019; Niu et al., 2021).

Recently, with the development of Transformer (Dosovitskiy et al., 2020; Liu et al., 2021), some works (Liu et al., 2022; Song et al., 2022; Yan et al., 2023a; Tel et al., 2023) bring in self- and

cross- attention to alleviate the ghosting artifacts. For example, Liu *et al.* (Liu et al., 2022) propose HDR-Transformer, which embeds a local context extractor into SwinIR (Liang et al., 2021) basic block for jointly capturing global and local dependencies. Song *et al.* (Song et al., 2022) suggest selectively applying the transformer and CNN model to misaligned and aligned areas, respectively. However, both CNN- and Transformer-based methods require sufficient labeled data for training networks, while the data collection is time-consuming and laborious.

## 2.2 FEW-SHOT AND SELF-SUPERVISED HDR IMAGING WITH MULTI-EXPOSURE IMAGES

To alleviate the reliance on HDR ground-truths, few-shot and self-supervised HDR reconstruction methods have been explored. FSHDR (Prabhakar et al., 2021) combines unlabeled dynamic samples with few labeled samples to train a neural network, then leverages the model output of unlabeled samples as a pseudo-HDR to generate pseudo-LDR images. Ultimately the HDR reconstruction network is learned with synthetic pseudo-pairs. Nazarczuk *et al.* (Nazarczuk et al., 2022) select well-exposure LDR patches as pseudo-HDR to generate pseudo-LDR, while the static LDR patches are directly merged for HDR ground-truths. However, due to unrealistic motion and illumination in synthetic LDR images, such methods exhibit performance gaps compared to supervised ones. Recently, SAME (Yan et al., 2023b) generates saturated regions in a self-supervised manner first, and then performs deghosting via a semi-supervised framework. But it still has limited performance improvement. In this work, we take full advantage of the internal characteristics of multi-exposure images to present a self-supervised approach SelfHDR, which achieves comparable performance to supervised ones.

Furthermore, some works incorporate emerging techniques to investigate self-supervised HDR reconstruction. For instance, GDP (Fei et al., 2023) exploits multi-exposure images to guide the denoising process of pre-trained diffusion generative models (Ho et al., 2020; Song et al., 2021), reconstructing HDR image. Mildenhall *et al.* (Mildenhall et al., 2022), Jun *et al.* (Jun-Seong et al., 2022), and Huang *et al.* (Huang et al., 2022a) employ NeRF (Mildenhall et al., 2020) to synthesize HDR images and the novel HDR views. However, these methods are less practical, since the specific models need to be re-optimized when facing new scenarios.

## 3 METHOD

### 3.1 MOTIVATION AND OVERVIEW

**Revisit Supervised HDR Reconstruction**. The combination of multi-exposure images enables HDR imaging in scenes with a wide range of brightness levels. In static scenes, the HDR image can be easily generated through a weighted sum of multi-exposure images (Debevec & Malik, 2008). However, when applying this approach in dynamic scenes, it will lead to ghosting artifacts. As a result, several recent works (Kalantari et al., 2017; Yan et al., 2019a; Liu et al., 2022; Tel et al., 2023) have suggested learning a deep neural network in a supervised manner for deghosting. Concretely, denote the LDR image taken with exposure time $t_i$ by $\boldsymbol{I}_i$, where $i = 1, 2, 3$ and $t_1 < t_2 < t_3$. They first map the LDR images to the linear domain, which can be written as,

$$\boldsymbol{H}_i = \boldsymbol{I}_i^{\gamma}/t_i, \tag{1}$$

where $\gamma$ denotes the gamma correction parameter and is generally set to 2.2. Then they concatenate $\boldsymbol{I}_i$ and $\boldsymbol{H}_i$, feeding them to the reconstruction network $\mathcal{R}$ with parameters $\Theta_{\mathcal{R}}$, *i.e.*,

$$\hat{\boldsymbol{Y}} = \mathcal{R}(\boldsymbol{X}_1, \boldsymbol{X}_2, \boldsymbol{X}_3; \Theta_{\mathcal{R}}), \tag{2}$$

where $\boldsymbol{X}_i = \{\boldsymbol{I}_i, \boldsymbol{H}_i\}$, $\hat{\boldsymbol{Y}}$ denotes the reconstructed HDR image. The optimized network parameters can be obtained by the following formula,

$$\Theta_{\mathcal{R}}^* = \arg\min_{\Theta_{\mathcal{R}}} \mathcal{L}(\mathcal{T}(\hat{\boldsymbol{Y}}), \mathcal{T}(\boldsymbol{Y})), \tag{3}$$

where $\mathcal{L}$ represents the loss function, $\boldsymbol{Y}$ denotes the HDR GT image. $\mathcal{T}$ is the tone-mapping process, represented as,

$$\mathcal{T}(\boldsymbol{Y}) = \frac{\log(1 + \mu\boldsymbol{Y})}{\log(1 + \mu)}, \text{ where } \mu = 5,000. \tag{4}$$

**Motivation of SelfHDR**. The acquisition of labeled data for HDR reconstruction is usually time-consuming and laborious. To alleviate the requirement of HDR GT, some works (Prabhakar et al., 2021; Yan et al., 2023b; Nazarczuk et al., 2022) have explored few-shot and zero-shot HDR reconstruction by constructing pseudo-pairs. However, their performance is unsatisfactory due to the gaps between the simulated pairs and real-world ones, especially in a fully self-supervised manner.

In this work, we expect to get rid of the demand for synthetic data, achieving self-supervised HDR reconstruction directly with real-world dynamic multi-exposure images. The goal should be feasible, as the multi-exposure images have provided probably sufficient information for HDR reconstruction. The property will be more intuitive when the color and structure are observed, respectively. On the one hand, the color of HDR images can be estimated from aligned inputs. On the other hand, the structure information of the HDR images can be generally discovered in some one of multi-exposure images, *i.e.*, most textures exist in the medium-exposure image, dark details are obvious in the high-exposure one, and bright scenes are clearly visible in the low-exposure one.

What we need to do is to dig for the right information from the multi-exposure images for constructing the HDR image. However, it is actually difficult to explore a straightforward self-supervised method that generates HDR images directly. Considering the above properties of HDR color and structure, we treat the two components respectively for ease of self-supervised implementation. Note that it can be a focus or emphasis on color and structure relatively, not necessarily an absolute separation.

Specifically, when training a self-supervised HDR reconstruction network with given multi-exposure images as input, suitable supervision signals have to be prepared. Instead of looking for a complete HDR image, we construct the color and structure components of the supervision respectively (see Sec. 3.2). Then we learn the network under the guidance of both components (see Sec. 3.3).

## 3.2 Constructing Color and Structure Components

### 3.2.1 Constructing Color Component

The color component should represent the HDR color as faithfully as possible, and it can be estimated by fusing the aligned multi-exposure images. Multiple frames in dynamic scenes are generally not aligned caused by camera shake or object motion. Although sometimes a few regions are aligned well, they are not enough to generate acceptable color components. In view

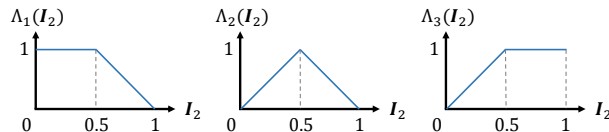

Figure 1: The triangle function that we use as the blending weights to generate color components.

of the effective capabilities of the optical flow estimation method (Liu et al., 2009), it is a natural idea to perform pre-alignment first. Concretely, taking the medium-exposure image $I_2$ as the reference, we calculate the optical flow from $I_2$ to $I_1$ and $I_3$, respectively. Thus, we can back warp $H_1$ and $H_3$ according to the calculated optical flow, obtaining $\tilde{H}_1$ and $\tilde{H}_3$ that are roughly aligned with $H_2$. Then we can predict the color component $Y_{color}$ with the following formula,

$$Y_{color} = \frac{A_1 \tilde{H}_1 + A_2 H_2 + A_3 \tilde{H}_3}{A_1 + A_2 + A_3}, \tag{5}$$

where $A_i$ represents pixel-level fusion weight. We follow Kalantari *et al.* (Kalantari et al., 2017) and express $A_i$ as,

$$A_1 = 1 - \Lambda_1(I_2), \quad A_2 = \Lambda_2(I_2), \quad A_3 = 1 - \Lambda_3(I_2), \tag{6}$$

where $\Lambda_i(I_2)$ is shown in Fig. 1.

When the images are perfectly aligned, the color components $Y_{color}$ can be regarded as an HDR image directly. However, such an ideal state is hard to reach due to object occlusion and sometimes non-robust optical flow model. Small errors during pre-alignment may cause blurring, while large ones cause ghosting in color components. Nevertheless, regardless of the ghosting areas, the rest can record the rough color value, and in which well-aligned ones can offer both good color and structure cues of HDR images. Moreover, for the areas with alignment errors, we further construct structure components to guide the reconstruction network in the next subsection.

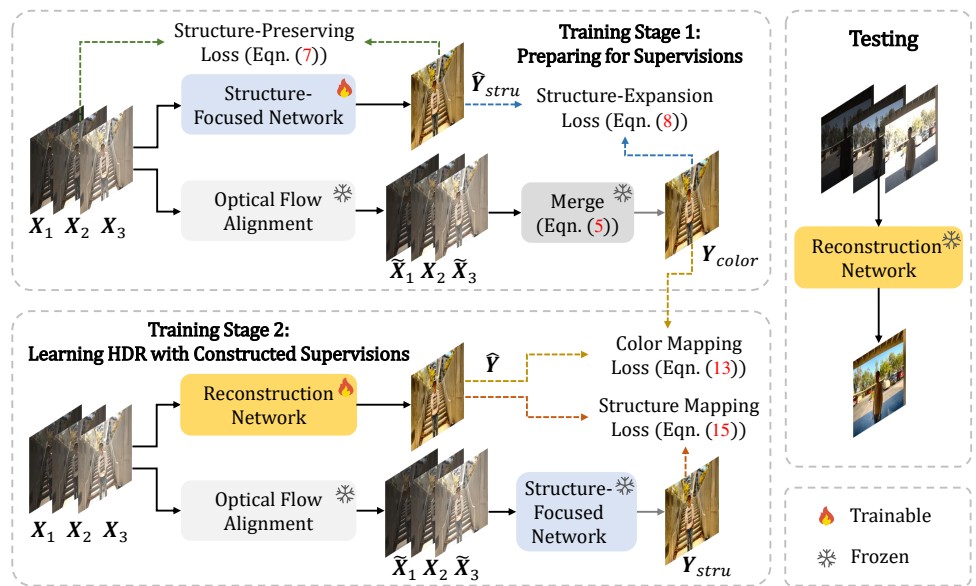

Figure 2: Overview of SelfHDR. During training, we first construct color and structure components (*i.e.*, $Y_{color}$ and $Y_{stru}$), then take $Y_{color}$ and $Y_{stru}$ for supervising the HDR reconstruction network. During testing, the HDR reconstruction network can be used to predict HDR images from unseen multi-exposure images. Dotted lines with different colors represent different loss terms.

### 3.2.2 CONSTRUCTING STRUCTURE COMPONENT

Although the medium-exposure image can provide most of the texture information, it is not optimal to take it as the only structure guidance for the HDR reconstruction, as the dark areas may be unclear and over-exposed ones may exist in it. Besides, it is not easy to put into practice when using the low-exposure and high-exposure images as guidance, due to the position and color differences between the HDR image and them. Fortunately, the previously constructed color component $Y_{color}$ can preserve the structure of dark and over-exposed areas to some extent. Therefore, we can combine the medium-exposure image and color component $Y_{color}$ to help construct the structure component.

Concretely, we first learn a structure-focused network with the guidance of medium-exposure image and color component $Y_{color}$. During training, the network takes the multi-exposure images as input, as shown in Fig. 2. On the one hand, the medium-exposure image guides the network to preserve well-exposed textures from the input reference image. It is accomplished by a structure-preserving loss $\mathcal{L}_{sp}$, which can be written as,

$$\mathcal{L}_{sp}(\hat{Y}_{stru}, H_2) = \|(\mathcal{T}(\hat{Y}_{stru}) - \mathcal{T}(H_2)) * M_{sp}\|_1, \tag{7}$$

where $\hat{Y}_{stru}$ denotes the network output. $M_{sp}$ emphasizes the well-exposed areas, and mitigates the adverse effects of dark and over-exposed ones in the reference image $H_2$. The function $\Lambda_2(I_2)$ (see Fig. 1) can do just that, so we set $M_{sp}$ to $\Lambda_2(I_2)$. On the other hand, the color component $Y_{color}$ guides the network to learn the structure from non-reference images by calculating structure-expansion loss $\mathcal{L}_{se}$, which can be written as,

$$\mathcal{L}_{se}(\hat{Y}_{stru}, Y_{color}) = \|(\mathcal{T}(\hat{Y}_{stru}) - \mathcal{T}(Y_{color})) * M_{se}\|_1. \tag{8}$$

$M_{se}$ is a binary mask, distinguishing whether the pixel of $Y_{color}$ is composited from well-aligned multi-exposure ones. We design each pixel $M_{se}^p$ of $M_{se}$ as,

$$M_{se}^p = \begin{cases} 1 & |((\mathcal{T}(Y_{color}) - \mathcal{T}(H_2)) * \Lambda_2(I_2))^p| < \sigma_{se} \\ 0 & |((\mathcal{T}(Y_{color}) - \mathcal{T}(H_2)) * \Lambda_2(I_2))^p| \geq \sigma_{se} \end{cases}, \tag{9}$$

where $\sigma_{se}$ is a threshold and set to $5/255$. In short, the parameter $\Theta_{\mathcal{S}}$ of structure-focused network $\mathcal{S}$ is jointly optimized by structure-preserving and structure-expansion loss terms, *i.e.*,

$$\Theta_{\mathcal{S}}^* = \arg\min_{\Theta_{\mathcal{S}}}[\mathcal{L}_{se}(\hat{Y}_{stru}, Y_{color}) + \lambda_{sp}\mathcal{L}_{sp}(\hat{Y}_{stru}, H_2)], \tag{10}$$

where $\lambda_{sp}$ denotes the weight coefficient of structure-preserving loss and is set to 4.

Then, we feed aligned multi-exposure images rather than original ones into the pre-trained structure-focused network $\mathcal{S}$. The final structure component $\boldsymbol{Y}_{stru}$ can be expressed as,

$$\boldsymbol{Y}_{stru} = \mathcal{S}(\tilde{\boldsymbol{X}}_1, \boldsymbol{X}_2, \tilde{\boldsymbol{X}}_3; \Theta_{\mathcal{S}}^*), \tag{11}$$

where $\tilde{\boldsymbol{X}}_1$ and $\tilde{\boldsymbol{X}}_3$ denote aligned $\boldsymbol{X}_1$ and $\boldsymbol{X}_3$ with the reference of $\boldsymbol{X}_2$. Such an operation can help structure-focused networks reduce the alignment burden, thus further enhancing the structure component. In addition, benefiting from the supervision of the color component, the structural component $\boldsymbol{Y}_{stru}$ also has some color cues, although it mainly focuses on the HDR textures.

## 3.3 LEARNING HDR WITH COLOR AND STRUCTURE COMPONENTS

With the color and structure components as guidance, we can train an HDR reconstruction network $\mathcal{R}$ without other ground-truths. The optimized network parameters $\Theta_{\mathcal{R}}^*$ can be modified from Eqn. (3) to the following formula,

$$\Theta_{\mathcal{R}}^* = \arg\min_{\Theta_{\mathcal{R}}}[\mathcal{L}_{color}(\hat{\boldsymbol{Y}}, \boldsymbol{Y}_{color}) + \lambda_{stru}\mathcal{L}_{stru}(\hat{\boldsymbol{Y}}, \boldsymbol{Y}_{stru})], \tag{12}$$

where $\hat{\boldsymbol{Y}}$ represents the network output. $\mathcal{L}_{color}$ and $\mathcal{L}_{stru}$ denote color mapping and structure mapping loss terms, respectively. $\lambda_{stru}$ is the weight coefficient of $\mathcal{L}_{stru}$ and is set to 1.

For color mapping term, we adopt $\ell_1$ loss, which can be written as,

$$\mathcal{L}_{color}(\hat{\boldsymbol{Y}}, \boldsymbol{Y}_{color}) = \|(\mathcal{T}(\hat{\boldsymbol{Y}}) - \mathcal{T}(\boldsymbol{Y}_{color})) * \boldsymbol{M}_{color}\|_1, \tag{13}$$

where $\boldsymbol{M}_{color}$ is similar as $\boldsymbol{M}_{se}$, and is also a binary mask. It excludes areas where optical flow is estimated incorrectly when generating $\boldsymbol{Y}_{color}$. Instead of using Eqn. (9), here we can utilize $\boldsymbol{Y}_{stru}$ to design a more accurate mask, which can be expressed as,

$$\boldsymbol{M}_{color}^p = \begin{cases} 1 & |(\mathcal{T}(\boldsymbol{Y}_{color}) - \mathcal{T}(\boldsymbol{Y}_{stru}))^p| < \sigma_{color} \\ 0 & |(\mathcal{T}(\boldsymbol{Y}_{color}) - \mathcal{T}(\boldsymbol{Y}_{stru}))^p| \geq \sigma_{color} \end{cases}, \tag{14}$$

where $p$ denotes a pixel, $\sigma_{color}$ is a threshold and set to $10/255$. For structure mapping term, we adopt VGG-based (Simonyan & Zisserman, 2015) perceptual loss, which can be written as,

$$\mathcal{L}_{stru}(\hat{\boldsymbol{Y}}, \boldsymbol{Y}_{stru}) = \sum_k \|\phi_k(\mathcal{T}(\hat{\boldsymbol{Y}})) - \phi_k(\mathcal{T}(\boldsymbol{Y}_{stru}))\|_1, \tag{15}$$

where $\phi_k(\cdot)$ denotes the output of $k$-th layer in VGG (Simonyan & Zisserman, 2015) network.

## 4 EXPERIMENTS

### 4.1 IMPLEMENTATION DETAILS

**Framework Details.** Note that this work does not focus on the design of network architectures, and we employ existing ones directly. The structure-focused and reconstruction networks use the same architecture. And we adopt CNN-based (*i.e.*, AHDRNet (Yan et al., 2019a) and FSHDR (Prabhakar et al., 2021)) and Transformer-based (*i.e.*, HDR-Transformer (Liu et al., 2022), and SCTNet (Tel et al., 2023)) networks for experiments, respectively. Besides, the optical flow is calculated by Liu *et al.* (Liu et al., 2009)'s approach, as recommended in (Kalantari et al., 2017; Prabhakar et al., 2021).

**Datasets.** Experiments are mainly conducted on Kalantari *et al.* dataset (Kalantari et al., 2017), which are extensively utilized in previous works. The dataset consists of 74 samples for training and 15 for testing. Each sample comprises three LDR images, captured at exposure values of $\{-2, 0, 2\}$ or $\{-3, 0, 3\}$, alongside a corresponding HDR GT image. We use these testing images for both quantitative and qualitative evaluations. Additionally, following (Kalantari et al., 2017; Yan et al., 2019a; Liu et al., 2022), we take the Sen *et al.* (Sen et al., 2012) and Tursun *et al.* (Tursun et al., 2016) datasets (without GT) for further qualitative comparisons.

**Training Details.** The structure-focused and reconstruction networks are trained successively, and share the same settings. The training patches of size $128 \times 128$ are randomly cropped from the original images. The batch size is set to 16. Adam (Kingma & Ba, 2015) with $\beta_1 = 0.9$ and $\beta_2 = 0.999$ is

Table 1: Quantitative results on Kalantari *et al.* dataset (Kalantari et al., 2017). 'SelfHDR$_{network}$' denotes the reconstruction network we use, *i.e.*, AHDRNet (Yan et al., 2019a), FSHDR (Prabhakar et al., 2021), HDR-Transformer (Liu et al., 2022), and SCTNet (Tel et al., 2023). The best results in each category are **bolded**.

| | Method | PSNR-$u$ / SSIM-$u$ | PSNR-$l$ / SSIM-$l$ | HDR-VDP-2 |
|---|---|---|---|---|
| Fully-Supervised | AHDRNet (*CVPR* 2019) | 43.63 / 0.9900 | 41.14 / 0.9702 | 64.61 |
| | FSHDR (*CVPR* 2021) | 43.03 / 0.9902 | **42.27** / 0.9889 | **64.79** |
| | HDR-Transformer (*ECCV* 2022) | 44.21 / **0.9918** | 42.17 / 0.9889 | 64.63 |
| | SCTNet (*ICCV* 2023) | **44.48** / 0.9916 | 42.00 / **0.9897** | 64.47 |
| Few (K)-Shot | FSHDR$_{K=5}$ (*CVPR* 2021) | **43.02** / 0.9874 | **41.98** / 0.9885 | **64.54** |
| | FSHDR$_{K=1}$ (*CVPR* 2021) | 42.52 / 0.9846 | 41.92 / **0.9887** | 64.41 |
| Self-Supervised | FSHDR$_{K=0}$ (*CVPR* 2021) | 42.17 / 0.9828 | 41.47 / 0.9884 | 64.21 |
| | Nazarczuk *et al.* (*ArXiv* 2022) | 42.15 / - | 40.54 / - | 63.99 |
| | Our SelfHDR$_{AHDRNet}$ | 43.68 / 0.9901 | 41.09 / 0.9873 | 64.57 |
| | Our SelfHDR$_{FSHDR}$ | 43.80 / 0.9902 | 41.72 / 0.9880 | 64.57 |
| | Our SelfHDR$_{HDR-Transformer}$ | 43.94 / **0.9907** | **41.79** / 0.9883 | **64.98** |
| | Our SelfHDR$_{SCTNet}$ | **43.95** / **0.9907** | 41.77 / **0.9889** | 64.77 |

taken to optimize models for 150 epochs. The learning rate is initially set to $1 \times 10^{-4}$ for CNN-based networks and $2 \times 10^{-4}$ for Transformer-based ones, and reduces by half every 50 epochs.

**Evaluation Configurations.** We use PSNR and SSIM (Wang et al., 2004) as evaluation metrics. PSNR and SSIM are both calculated on the linear and tone-mapped domains, denoted as '-$l$' and '-$u$', respectively. Moreover, we adopt HDR-VDP-2 (Mantiuk et al., 2011) that measures the human visual difference between results and targets. The higher HDR-VDP-2, the better results.

## 4.2 COMPARISON WITH STATE-OF-THE-ARTS

As described in Sec. 4.1, we adopt four existing HDR reconstruction networks (*i.e.*, AHDRNet, FSHDR, HDR-Transformer, and SCTNet) for experiments, respectively. We compare them with the corresponding supervised manners and two self-supervised methods (*i.e.*, FSHDR$_{K=0}$ and Nazarczuk *et al.* (Nazarczuk et al., 2022)). And the results of few-shot FSHDR are also provided.

**Quantitative Results.** Table 1 shows the quantitative comparison results. As loss functions are always calculated on tone-mapped images, and HDR images are typically viewed on LDR displays, we suggest taking evaluation metrics in the tone-mapped domain (*i.e.*, PSNR-$u$ and SSIM-$u$) as the primary reference. From the table, four SelfHDR versions all outperform the previous self-supervised methods. Especially, with the same reconstruction network, our SelfHDR$_{FSHDR}$ achieves 1.58 dB PSNR gain than FSHDR$_{K=0}$. The results of SelfHDR can be further improved with the use of more advanced reconstruction networks (*i.e.*, HDR-Transformer and SCTNet). Moreover, in comparison with the corresponding supervised methods, SelfHDR has comparable performance overall.

**Qualitative Results.** The visual comparisons on Kalantari *et al.* dataset (Kalantari et al., 2017) as well as Sen *et al.* (Sen et al., 2012) and Tursun *et al.* (Tursun et al., 2016) datasets are shown in Fig. 3 and Fig. 4, respectively. Our results have fewer artifacts than FSHDR$_{K=0}$, and sometimes even outperform the corresponding supervised methods. They show the same trend as the quantitative ones. Please see Sec. E in the appendix for more results.

## 5 ABLATION STUDY

The ablation studies are all conducted using AHDRNet (Yan et al., 2019a) as the structure-focused and reconstruction network.

## 5.1 EFFECT OF COLOR AND STRUCTURE SUPERVISION

The quantitative results of color and structure components ($Y_{color}$ and $Y_{stru}$) are given in Table 2. From the table, the final HDR results achieve better performance than both $Y_{color}$ and $Y_{stru}$ on PSNR-$u$, SSIM-$u$, and HDR-VDP-2. It indicates that the two components are complementary and

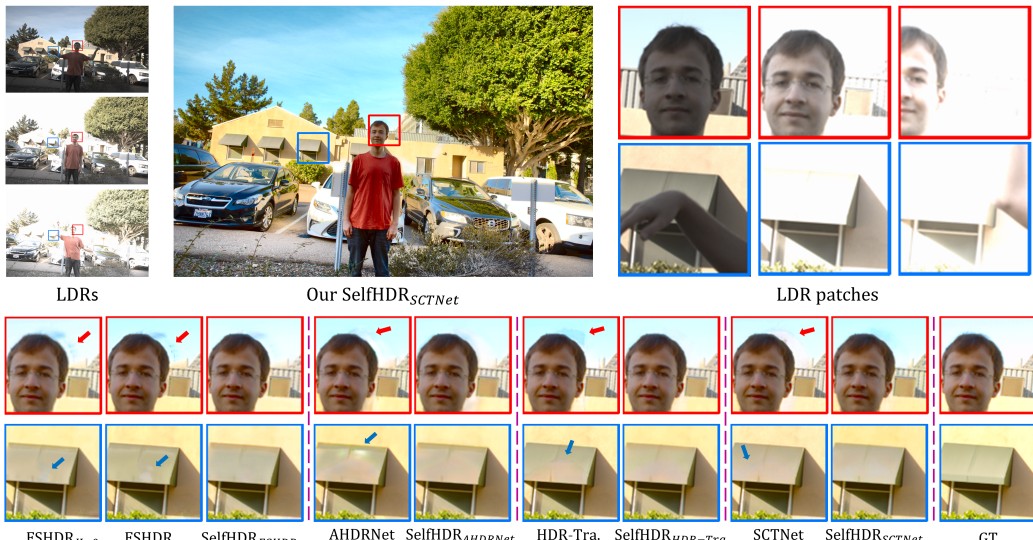

Figure 3: Visual comparison on Kalantari *et al*. dataset ([Kalantari et al., 2017](#)). Red and blue arrows indicate areas with ghosting artifacts from other methods. 'HDR-Tra.' denotes HDR-Transformer.

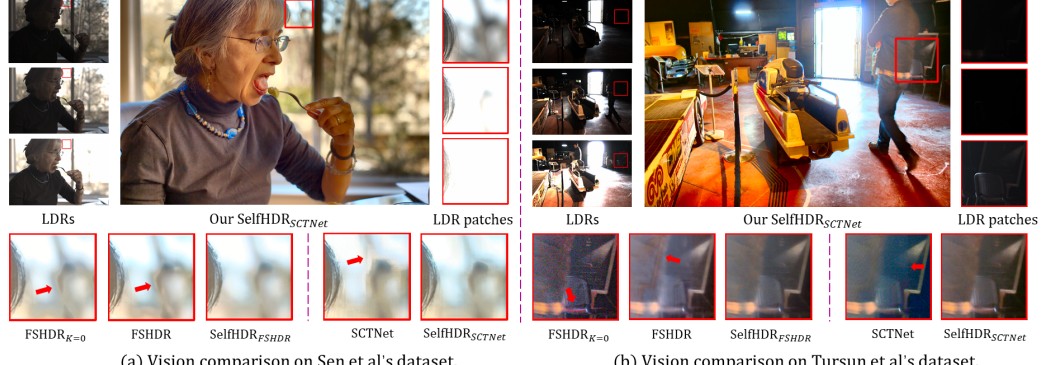

Figure 4: Visual comparison on (a) Sen *et al*. ([Sen et al., 2012](#)) and (b) Tursun *et al*. ([Tursun et al., 2016](#)) datasets. Red arrows indicate areas with poor quality from other methods.

Table 2: Quantitative results of supervision components and final HDR images.

|  | PSNR-$u$ / SSIM-$u$ | PSNR-$l$ / SSIM-$l$ | HDR-VDP-2 |
|---|---|---|---|
| Color Components $\boldsymbol{Y}_{color}$ | 34.45 / 0.9652 | 39.01 / 0.9783 | 58.28 |
| Structure Components $\boldsymbol{Y}_{stru}$ | 43.38 / 0.9891 | 41.74 / 0.9874 | 64.48 |
| Final HDR Images | 43.68 / 0.9901 | 41.09 / 0.9873 | 64.57 |

taking them as supervision is appropriate and effective. Furthermore, we conduct the following two experiments to further illustrate the effectiveness.

**Comparision with Component Fusion.** It may be a more natural idea to obtain HDR results by fusing the color and structure components. Here we implement that by calculating $\boldsymbol{M}_{color}\boldsymbol{Y}_{color} + (1 - \boldsymbol{M}_{color})\boldsymbol{Y}_{stru}$. We empirically re-adjust the hyperparameter $\sigma_{color}$ in Eqn. (14), but find it gets the best results when $\boldsymbol{M}_{color} = \boldsymbol{0}$. In other words, it is difficult to achieve better results by fusing two components simply. Instead, our SelfHDR provides a more flexible and efficient way.

**Refining Structure Component.** Denote $\hat{\boldsymbol{Y}}^*$ by the reconstruction network output when inputting multi-exposure images aligned by optical flow ([Liu et al., 2009](#)). From another point of view, $\hat{\boldsymbol{Y}}^*$ can be regarded as a refined structure component with higher quality. Thus, we further take $\boldsymbol{Y}_{color}$ and $\hat{\boldsymbol{Y}}^*$ as new supervisions to re-train a reconstruction model, while the performance does not improve. It demonstrates that $\boldsymbol{Y}_{stru}$ generated by structure-focused network is already sufficient.

Table 3: Effect of loss terms ($\mathcal{L}_{se}$ and $\mathcal{L}_{sp}$) when training structure-focused network.

| $\mathcal{L}_{se}$ / $\mathcal{L}_{sp}$ | $\boldsymbol{Y}_{stru}$ PSNR-$u$ / PSNR-$l$ | $\hat{\boldsymbol{Y}}$ PSNR-$u$ / PSNR-$l$ |
|---|---|---|
| × / ✓ | 38.24 / 33.61 | 38.79 / 33.72 |
| ✓ / × | 42.69 / **41.89** | 43.09 / **41.13** |
| ✓ / ✓ | **43.38** / 41.74 | **43.68** / 41.09 |

Table 4: Effect of different $\boldsymbol{M}_{color}$ when training reconstruction network.

| $\boldsymbol{M}_{color}$ | $\hat{\boldsymbol{Y}}$ PSNR-$u$ / PSNR-$l$ |
|---|---|
| × | 43.55 / **41.12** |
| Eqn. (9) | 43.59 / 41.02 |
| Eqn. (14) | **43.68** / 41.09 |

Table 5: Effect of the designed masks ($\boldsymbol{M}_{se}$ and $\boldsymbol{M}_{sp}$) when training structure-focused network.

| $\boldsymbol{M}_{se}$ / $\boldsymbol{M}_{sp}$ | $\boldsymbol{Y}_{stru}$ PSNR-$u$ / PSNR-$l$ | $\hat{\boldsymbol{Y}}$ PSNR-$u$ / PSNR-$l$ |
|---|---|---|
| × / × | 38.26 / 33.68 | 38.82 / 33.71 |
| ✓ / × | 38.29 / 33.65 | 38.86 / 33.82 |
| × / ✓ | 43.26 / 41.73 | 43.60 / 41.07 |
| ✓ / ✓ | **43.38** / **41.74** | **43.68** / **41.09** |

Table 6: Effect of pre-alignment processing when constructing $\boldsymbol{Y}_{color}$ and $\boldsymbol{Y}_{stru}$.

| $\boldsymbol{Y}_{color}$ / $\boldsymbol{Y}_{stru}$ | $\hat{\boldsymbol{Y}}$ PSNR-$u$ / PSNR-$l$ |
|---|---|
| × / × | 35.50 / 34.95 |
| × / ✓ | 41.66 / 40.90 |
| ✓ / × | 43.41 / 40.76 |
| ✓ / ✓ | **43.68** / **41.09** |

## 5.2 EFFECT OF LOSS TERMS AND MASKS

**Structure-Focused Network.** The structure-focused network is trained with the supervision of color component and input reference, implementing by calculating structure-expansion loss $\mathcal{L}_{se}$ and structure-preserving loss $\mathcal{L}_{sp}$, respectively. Here we explore the effect of different supervisions by using $\mathcal{L}_{sp}$ or $\mathcal{L}_{se}$ only. From Table 3, it can be seen that $\mathcal{L}_{sp}$ may play a weaker role, as it mainly constrains the well-exposed areas whose structure may be also fine in $\boldsymbol{Y}_{color}$. Nevertheless, combining two supervisions is more favorable than using one, thus both are indispensable.

Moreover, we conduct ablation experiments on the designed masks ($\boldsymbol{M}_{sp}$ and $\boldsymbol{M}_{se}$) in loss terms. The results in Table 5 show that the masks are competent in avoiding harmful information from supervision. The visualizations of the masks are given in Sec. A of the appendix.

**Reconstruction Network.** For training the reconstruction network, the adverse effect of ghosting regions from color supervision $\boldsymbol{Y}_{color}$ needs to be avoided as well. We utilize structure component to design a more accurate mask in Eqn. (14), and it does show better results than Eqn. (9) from Table 4.

In addition, we conduct the experiments with different hyperparameters $\sigma_{se}$ (see Eqn. (9)) and $\sigma_{color}$ (see Eqn. (14)) in Sec. D of the appendix.

## 5.3 EFFECT OF OPTICAL FLOW PRE-ALIGNMENT

When constructing color and structure supervisions, the inputs need to be pre-aligned by the optical flow approach. Here we remove the pre-alignment processing separately to investigate its impact on the final HDR results, which are shown in Table 6. From the table, pre-alignment during obtaining $\boldsymbol{Y}_{color}$ is crucial, and pre-alignment during obtaining $\boldsymbol{Y}_{stru}$ can further improve performance. The corresponding quantitative results of $\boldsymbol{Y}_{color}$ and $\boldsymbol{Y}_{stru}$ can be seen in Sec. B of the appendix.

## 6 CONCLUSION

By exploiting the internal properties of multi-exposure images, we propose a self-supervised high dynamic range (HDR) reconstruction method named SelfHDR for dynamic scenes. In SelfHDR, the reconstruction network is learned under the supervision of two complementary components, which focus on the color and structure of HDR images, respectively. The color components are synthesized by merging aligned multi-exposure images. The structure components are constructed by feeding aligned inputs into the structure-focused network, which is trained with the supervision of color components and input reference images. Experiments show that SelfHDR outperforms the state-of-the-art self-supervised methods, and achieves comparable results to supervised ones. The discussion on method limitation and future work can be seen in Sec. F and Sec. G of the appendix.

## ACKNOWLEDGMENTS

This work is partially supported by the National Natural Science Foundation of China (NSFC) under Grant No. U19A2073.

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

APPENDIX

The content of the appendix material involves:

## A  ANALYSIS AND VISUALIZATION OF MASKS

In order to avoid harmful information in supervision during training structure-focused network, we carefully design the mask $M_{sp}$ and $M_{se}$ for calculating structure-preserving loss $\mathcal{L}_{sp}$ and structure-expansion loss $\mathcal{L}_{se}$, respectively. The quantitative results of related ablation experiments are shown in Table 5. Here we give more analysis about the elaborate masks and visualize an example in Fig. A. Therein, the corresponding color component is shown in Fig. A (g).

**Mask $M_{sp}$ in Structure-Preserving Loss.** The structure-preserving loss aims at guiding the network to preserve textures of the input reference image, and it is calculated between model output and linear medium-exposure image $H_2$. From Table 5, it leads to poor performance when measuring the distance between the two directly, as the structural information of dark and over-exposed regions is incomplete in medium-exposure image $H_2$.

Thus, we suggest embedding a mask $M_{sp}$ into the loss, and it should emphasize the well-exposed areas and mitigate the adverse effects of dark as well as over-exposed areas. $\Lambda_2(I_2)$ in Fig. 1 can do just that, and we adopt it as $M_{sp}$. The visualization of a $\Lambda_2(I_2)$ example is shown in Fig. A (i). It can be seen that the overexposed area surrounded by the blue box is successfully suppressed.

**Mask $M_{se}$ in Structure-Expansion Loss.** The structure-expansion loss aims at guiding the network to learn textures from non-reference inputs, and it is calculated between model output and color component $Y_{color}$. As $Y_{color}$ is obtained by fusing aligned multi-exposure images (see Fig. A (b), (e), and (f)), it is inevitable that ghosting artifacts exist in $Y_{color}$ (see the area surrounded by the red box in Fig. A (g)) when the alignment fails.

Thus, a mask $M_{se}$ should be designed to circumvent the adverse effects of these ghosting areas for better guiding the network. It is not appropriate to calculate the mask based on the simple difference between $Y_{color}$ and reference image $H_2$. Because even if the dark and over-exposed areas are aligned well, the difference between $Y_{color}$ and $H_2$ is still large (see the area surrounded by the blue box in Fig. A (j)). As a result, we utilize $\Lambda_2(I_2)$ again to mitigate the adverse effects of these areas. Specifically, we multiply $\Lambda_2(I_2)$ to $Y_{color}$ and $H_2$ for calculating the difference, as shown in Eqn. (9). A mask example is shown in Fig. A (k). It can be seen that the well-aligned over-exposed areas surrounded by the blue box are successfully retained, and only the misaligned area is masked.

With the designed masks, the generated structure component $Y_{stru}$ in Fig. A (l) combines the strengths of the supervisions $Y_{color}$ and $H_2$, while discarding their weaknesses.

## B  EFFECT OF OPTICAL FLOW PRE-ALIGNMENT

When constructing color and structure supervisions, the inputs need to be pre-aligned by the optical flow approach (Liu et al., 2009). We remove the pre-alignment processing separately to investigate its impact on the final HDR results, which are shown in Table A. From the table, the pre-alignment during obtaining $Y_{color}$ is crucial, as $Y_{color}$ affects the quality of $Y_{stru}$, while $Y_{color}$ and $Y_{stru}$ decide the final HDR result. On this basis, pre-alignment during generating $Y_{stru}$ can further improve performance, achieving 0.27 dB PSNR gain on the HDR result $\hat{Y}$.

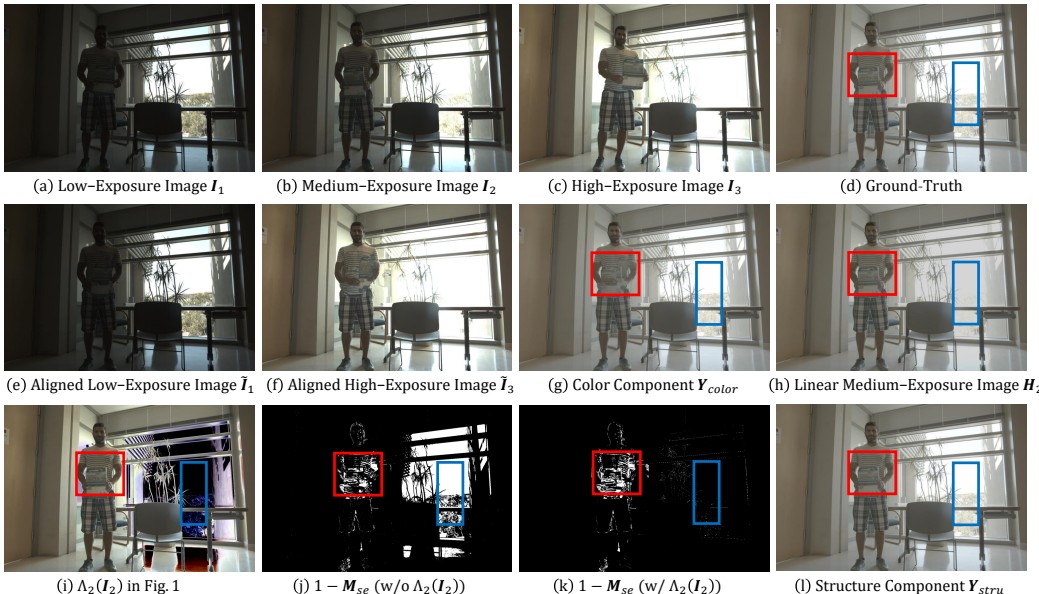

(a) Low–Exposure Image $I_1$    (b) Medium–Exposure Image $I_2$    (c) High–Exposure Image $I_3$    (d) Ground-Truth

(e) Aligned Low–Exposure Image $\bar{I}_1$    (f) Aligned High–Exposure Image $\bar{I}_3$    (g) Color Component $Y_{color}$    (h) Linear Medium–Exposure Image $H_2$

(i) $\Lambda_2(I_2)$ in Fig. 1    (j) $1 - M_{se}$ (w/o $\Lambda_2(I_2)$)    (k) $1 - M_{se}$ (w/ $\Lambda_2(I_2)$)    (l) Structure Component $Y_{stru}$

Figure A: Visualization of masks and related images. (a)∼(c) show the multi-exposure images, while (d) is the corresponding ground-truth from Kalantari *et al*. (Kalantari et al., 2017) dataset. (e) and (f) show the aligned low-exposure and aligned high-exposure images, respectively, which are obtained by optical flow (Liu et al., 2009) alignment with the reference of medium-exposure image. (g) is the constructed color component by fusing aligned multi-exposure images. (h) is the medium-exposure image in the linear domain. (i) is the mask as a blending weight in Fig. 1. (j) and (k) denote the masks $1 - M_{se}$ (see Eqn. (9)) constructed without and with $\Lambda_2(I_2)$, respectively. (l) is the generated structure component. The red box indicates the area where optical flow alignment fails, and the blue box indicates the area with high brightness.

Table A: Effect of pre-alignment processing when constructing supervision information ($Y_{color}$ and $Y_{stru}$). The final HDR results ($\hat{Y}$) are obtained by learning the model with corresponding ($Y_{color}$ and $Y_{stru}$) supervisions.

| | $Y_{color}$ | $Y_{stru}$ | $Y_{color}$ PSNR-$u$ / PSNR-$l$ | $Y_{stru}$ PSNR-$u$ / PSNR-$l$ | $\hat{Y}$ PSNR-$u$ / PSNR-$l$ |
|---|---|---|---|---|---|
| | $\times$ | $\times$ | 25.69 / 31.31 | 34.58 / 34.35 | 35.50 / 34.95 |
| Pre-Alignment | $\times$ | $\checkmark$ | 25.69 / 31.31 | 39.04 / 40.38 | 41.66 / 40.90 |
| Processing | $\checkmark$ | $\times$ | **34.45 / 39.01** | 43.07 / 40.45 | 43.41 / 40.76 |
| | $\checkmark$ | $\checkmark$ | **34.45 / 39.01** | **43.38 / 41.74** | **43.68 / 41.09** |

In addition, we further evaluate the effect of pre-alignment on generating $Y_{stru}$. Specifically, we test the structure-focused network on 74 training scenes with and without optical flow pre-alignment, respectively. The results of $Y_{stru}$ show the pre-alignment manner has 0.44dB PSNR-$u$ and 1.46dB PSNR-$l$ gains on average. Moreover, we compare the results between the two manners one by one. We find that only in 6 scenes, the results without pre-alignment are more than 0.1dB better than those with pre-alignment on PSNR-$u$. In the other 68 scenes, the pre-alignment manner always gives better or comparable results.

## C  EFFECT OF DIFFERENT ALIGNMENT METHODS

The quality of color components mainly relies on the alignment method. In this work, for the sake of fairness, we follow Kalantari *et al*. (Kalantari et al., 2017) and FSHDR (Prabhakar et al., 2021), adopting Liu *et al*. (Liu et al., 2009)'s approach for optical flow alignment. Besides, we additionally conduct experiments with other commonly used optical flow networks, *i.e.*, PWC-Net (Sun et al.,

Table B: Effect of optical flow alignment methods.

| Alignment Method | PSNR-$u$ / SSIM-$u$ | PSNR-$l$ / SSIM-$l$ | HDR-VDP-2 |
|---|---|---|---|
| PWC-Net (Sun et al., 2018) | 43.45 / 0.9898 | 40.67 / 0.9864 | 64.07 |
| FlowFormer (Huang et al., 2022b) | 43.50 / 0.9900 | 40.60 / 0.9862 | 64.43 |
| Liu *et al.* (Liu et al., 2009) | **43.68 / 0.9901** | **41.09 / 0.9873** | **64.57** |

Table C: Effect of $\sigma_{se}$ in Eqn. (9).

| $\sigma_{se}$ | $\boldsymbol{Y}_{stru}$ PSNR-$u$ / PSNR-$l$ | $\hat{\boldsymbol{Y}}$ PSNR-$u$ / PSNR-$l$ |
|---|---|---|
| 2.5/255 | 43.27 / 41.49 | 43.64 / 40.88 |
| 5/255 | **43.38 / 41.74** | **43.68 / 41.09** |
| 7.5/255 | 43.18 / 41.67 | 43.57 / 41.04 |

Table D: Effect of $\sigma_{color}$ in Eqn. (14).

| $\sigma_{color}$ | $\hat{\boldsymbol{Y}}$ PSNR-$u$ / PSNR-$l$ |
|---|---|
| 5/255 | 43.60 / 41.08 |
| 10/255 | **43.68** / 41.09 |
| 15/255 | 43.61 / **41.10** |

2018) and FlowFormer (Huang et al., 2022b). As shown in Tab. B, although Liu *et al.*'s approach is relatively early, it is more robust for multi-exposure image alignment than recent learning-based PWC-Net and FlowFormer.

## D  ABLATION STUDY ON ADJUSTING $\sigma_{se}$ AND $\sigma_{color}$

The hyperparameters $\sigma_{se}$ (see Eqn. (9)) and $\sigma_{color}$ (see Eqn. (14)) are set to $5/255$ and $10/255$ by default for experiments, respectively. Here, we vary $\sigma_{se}$ or $\sigma_{color}$ to conduct experiments. Table C and D show the experimental results, respectively. The results show that the sensitivity $\sigma_{se}$ and $\sigma_{color}$ of our SelfHDR is acceptable.

## E  ADDITIONAL QUALITATIVE COMPARISONS

Additional visual comparisons on Kalantari *et al.* (Kalantari et al., 2017) dataset are shown in Fig. B and Fig. C, respectively. Our SelfHDR has fewer ghosting artifacts than zero-shot FSHDR (*i.e.*, FSHDR$_{K=0}$) (Prabhakar et al., 2021). Sometimes, SelfHDR even outperforms the corresponding supervised methods. Red arrows in the results indicate areas with ghosting artifacts in other methods.

## F  LIMITATION

First, the main limitation is the requirement for clear input images, *i.e.*, they should be noise-free and blur-free. When noise exists in short-exposure images or blur exists in long-exposure images, SelfHDR can not remove noise and blur, as shown in Fig. D. Second, when the scene irradiance changes drastically in shooting multi-exposure images, SelfHDR may fail, as the constructed color components may be inaccurate.

Actually, most existing multi-exposure HDR reconstruction methods (including supervised and self-supervised ones) only focus on removing ghosting artifacts caused by misalignment between inputs, having these limitations as well. Our SelfHDR has taken a step toward more realistic self-supervised HDR imaging by deghosting, while our ongoing work is to further address these limitations.

## G  FUTURE WORK

In future work, we will explore self-supervised HDR reconstruction when considering more realistic shooting conditions. In low-light environments, there may exist noise in short-exposure images and blur in long-exposure images. In order to achieve a self-supervised algorithm, we can combine HDR reconstruction with some self-supervised denoising and deblurring works to process input images for removing undesirable degradations. Moreover, an adaptive method may need to be explored to select a more appropriate image as a base frame. For example, when a medium-exposure image

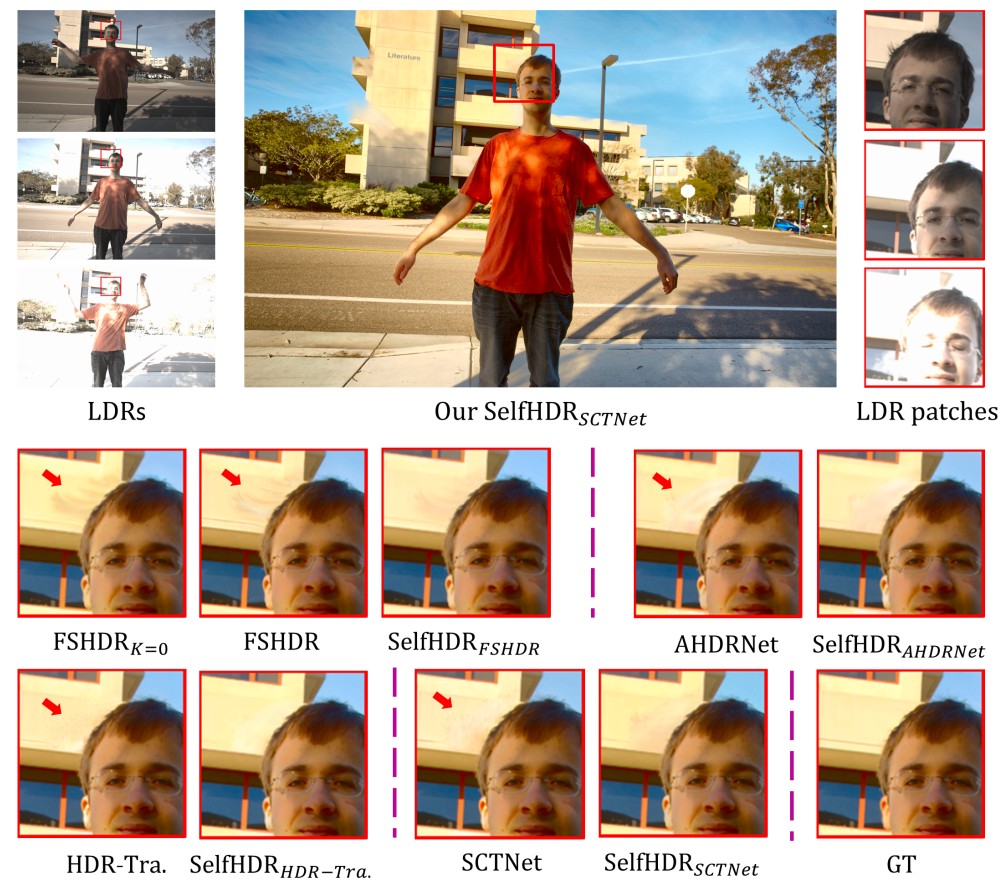

Figure B: Visual comparison on Kalantari *et al.* dataset (Kalantari et al., 2017). Red arrows indicate areas with ghosting artifacts from other methods. 'HDR-Tra.' denotes HDR-Transformer.

suffers more severe degradations than others, the method should adaptively take short-exposure or long-exposure images as a new base frame for HDR reconstruction.

In addition, as a self-supervised method, it has the potential to produce better results and bring better generalization by exploiting more multi-exposure images without the target HDR images. We will explore scaling up training data in the future.

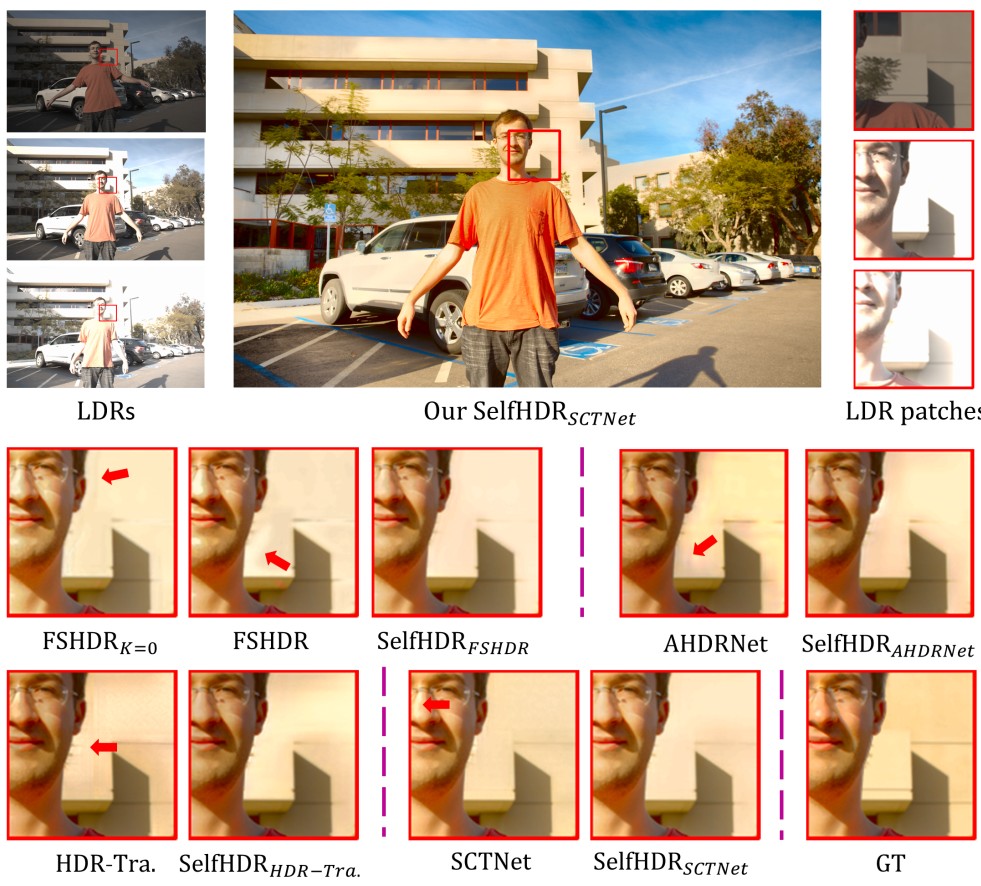

Figure C: Visual comparison on Kalantari *et al*. dataset (Kalantari et al., 2017). Red arrows indicate areas with ghosting artifacts from other methods. 'HDR-Tra.' denotes HDR-Transformer.

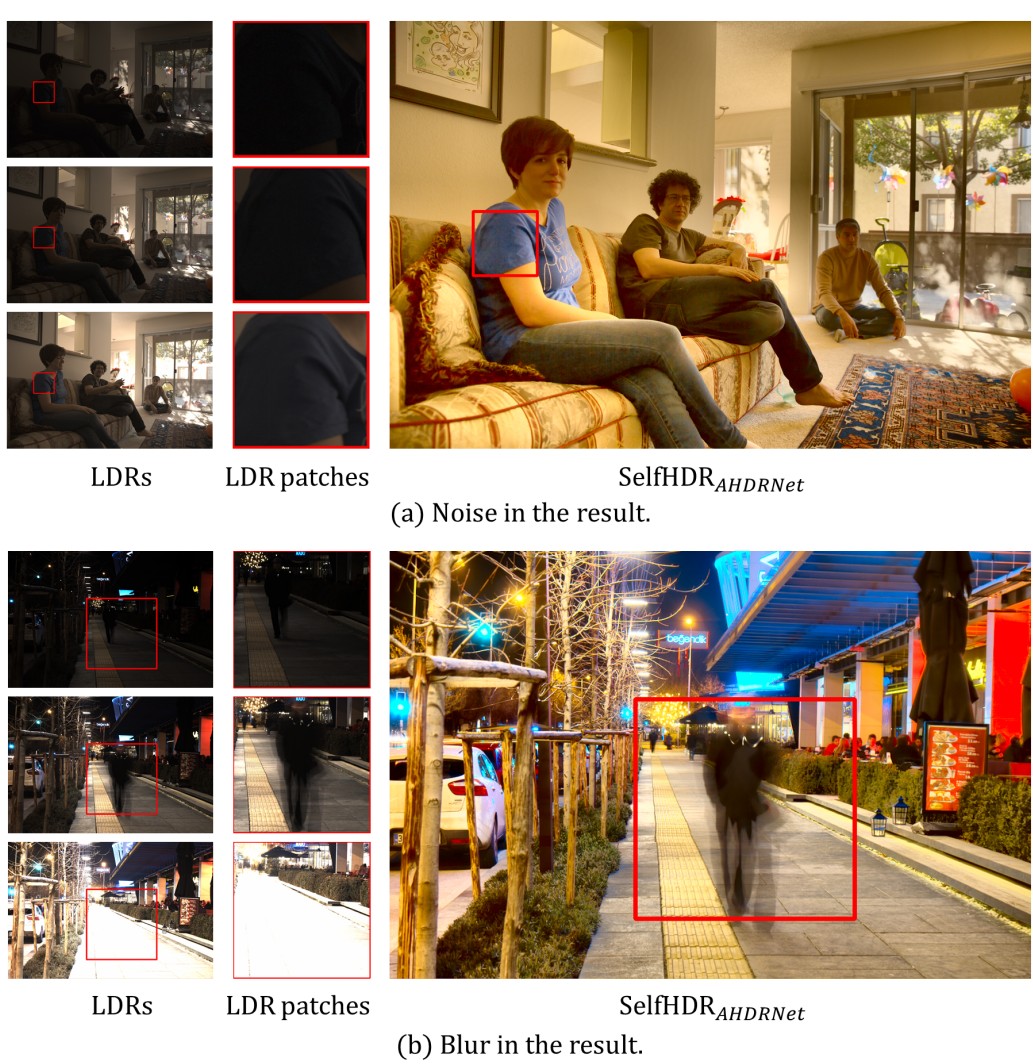

LDRs     LDR patches        SelfHDR$_{AHDRNet}$

(a) Noise in the result.

LDRs     LDR patches        SelfHDR$_{AHDRNet}$

(b) Blur in the result.

Figure D: Failure cases. Noise or blur may exist in the results.

