# OpenReview forum: "Self-Supervised High Dynamic Range Imaging with Multi-Exposure Images in Dynamic Scenes"
_ICLR.cc/2024/Conference — ICLR 2024 poster_

### Official Review · Reviewer_DQCr · 2023-10-21

**Soundness:** 3 good
**Presentation:** 3 good
**Contribution:** 3 good
**Rating:** 8
**Confidence:** 4

**Summary:**

The paper proposed a self-supervised HDR reconstruction method from a triplet of LDR images with relative motion. The proposed method first prepares two intermediate components focusing on HDR structure and color, and then learns a neural network to estimate the final HDR under the supervision of both. The approach is superior to other self-supervised learning approaches and is comparable to supervised learning methods with regard to objective metrics and visual qualities.

**Strengths:**

* The idea of using two-stage training to learn structure and color information first and then learn the final output is novel to me. The overall design of the method is reasonable.

* experiments are extensive and demonstrate the compelling performance of the proposed method.

**Weaknesses:**

* The proposed method is specific to 3 inputs. The 3 inputs should be informatively captured, especially the mid-exposed image, which should have texture and color of high fidelity as the reference. The requirement for the input is strict and can limit the application of the proposed method.

*The authors should point readers to the appendix when appropriate, e.g., visualization of masks and selection of sigmas.

**Questions:**

The structure-focused network was proposed to avoid errors caused by optical flow alignment, but it is integrated with optical flow alignment in training stage 2, which doesnt make sense to me. Although Tab. 6 shows improvement in general, I think it will hurt the performance in the cases when the alignment has more errors, whereas those cases can demonstrate the core value of the paper. The authors should elaborate on this and experiment on those cases to justify their statement when necessary.

Misc:
* HDR-VDP-2 metric is not as commonly used as PSNR/SSIM. The authors should briefly introduce what the value means, and if a bigger/smaller value indicates better image quality.
* In Tab 3-6 the highest scores can be marked in bold.

---

> ### Author Response · Authors · 2023-11-22
> **Response to Reviewer DQCr**
>
> We thank the reviewer for the valuable comments and suggestions.
> We appreciate the reviewer's questions and hope our responses could address the concerns.
>
>
> 1. Limitation of Inputs
>
> We agree with the reviewer's insight on method limitations. Moreover, most existing multi-exposure HDR reconstruction methods (including supervised and self-supervised ones) only focus on removing ghosting artifacts caused by misalignment between inputs, having this limitation as well. This work (SelfHDR) has taken a step toward more realistic self-supervised HDR imaging by deghosting, while our ongoing work is to further address this limitation.
>
>
> Here we share two possible solutions. On the one hand, we can combine HDR reconstruction with some self-supervised image restoration methods when considering more realistic shooting conditions. For example, there may exist noise in short-exposure images and blur in long-exposure images. In order to achieve a self-supervised algorithm, we can combine HDR reconstruction with some self-supervised denoising and debluring works to process input images for removing degradations.
>
> On the other hand, an adaptive method may need to be explored to select a more appropriate image as a base frame. For example, when a mid-exposure image suffers more severe degradations than others, the method should adaptively take short-exposure or long-exposure images as a new base frame for HDR reconstruction. Thus, it can alleviate the shooting requirements for a mid-exposure image.
>
> We will add this in Sec. E of the appendix in the revision.
>
>
> 2. Pre-Alignment in Structure-Focused Network
>
>
>
> The ultimate goal of the structure-focused network is to provide HDR supervision with good structure that is complementary to the color component. It should avoid ghosting results, rather than avoid errors caused by optical flow alignment. Moreover, different from naive merging operation (in Eqn. (5)) for generating color component, learning-based structure-focused network has the ability to handle misalignment problem, including the remaining misalignment after optical flow pre-alignment, as shown in Fig. A (l).
>
>
>
> Below we further explain the effect of pre-alignment. On the one hand, significantly incorrect optical flow mainly appears in occlusion and overexposed areas from other frames. These areas have little valuable information, so the results mainly rely on the base frame, and pre-alignment or not has little impact on the results. Thus, it's not bad to perform pre-alignment in this case. On the other hand, when optical flow is roughly correct, pre-alignment reduces the processing burden of the structure-focused network that does not adopt explicit alignment. Thus, it naturally brings improvements.
>
>
>
> We additionally test the structure-focused network on 74 training scenes with and without optical flow pre-alignment, respectively. First, the average results are shown in the following table. It can be seen that the pre-alignment manner has a significant improvement on average. Second, we compare the results between the two manners one by one. We find that only in 6 scenes, the results without pre-alignment are more than 0.1dB better than those with pre-alignment on PSNR-$u$. In the other 68 scenes, the pre-alignment manner always gives better or comparable results. Moreover, we also observe the characteristics of these 6 scenes. We find that their alignment is not worse than the other 68 scenes and is moderate in general. The experiments show that the above explanation is reasonable in most scenes.
>
>
>
> |  | PSNR-$u$ / SSIM-$u$ | PSNR-$l$ / SSIM-$l$
> | :----:       |  :----:  | :----:  |
> | w/o Pre-Alignment        | 42.40 / 0.9809    | 40.02 / 0.9869   |
> | w/ Pre-Alignment       | 42.84 / 0.9814    | 41.48 / 0.9894   |
>
> We will add this in Sec. B of the appendix in the revision.
>
>
> 3. Paper Writing and Misc
>
> Many thanks. In the revision, we will add pointers to the contents of the appendix. We will also briefly introduce HDR-VDP-2 and mark the best results in Tables 3$\sim$6 in bold.

---

> > ### Comment · Reviewer_DQCr · 2023-12-01
> >
> > The authors' responses address my concerns. I would like to keep my original rating.

---

### Official Review · Reviewer_TwnM · 2023-10-25

**Soundness:** 2 fair
**Presentation:** 2 fair
**Contribution:** 2 fair
**Rating:** 6
**Confidence:** 3

**Summary:**

This paper presents "SelfHDR," a self-supervised High Dynamic Range (HDR) image reconstruction method. Traditional HDR reconstruction methods require ample ground-truth data, which can be difficult and expensive to collect. The novelty in SelfHDR comes from learning a reconstruction network under two complementary components, HDR color and structure, constructed from multi-exposure static images. They designed a detailed process to generate these components and used them to train the reconstruction network. Experimental results showed that SelfHDR surpassed state-of-the-art self-supervised methods and achieved comparable performance with supervised methods. Contributions include a novel solution to alleviate the need for extensive labeled data for HDR image reconstruction.

**Strengths:**

The originality of the paper is substantial as it proposes an innovative self-supervised method for HDR image reconstruction, eliminating the need for extensive labeled data, a factor that typically hinders HDR reconstruction methods. The quality of the research is commendable. To build this method, they analyzed HDR images thoroughly and isolated the primary components, color, and structure that could be derived from multi-exposure static images. The paper is coherent and clear in its exposition, and the authors elegantly illustrate the problem, their approach, and results. The significance of this work is considerable as it presents a promising alternative for HDR image reconstruction, which demonstrates competitiveness with supervised methods while minimizing data requirements.

**Weaknesses:**

A potential weakness of the paper is the lack of a broader experimental validation. While they have tested the model on several datasets, the application in real-world scenarios, especially dealing with complex situations and variable lighting conditions, is not clearly examined. The paper could have also delved more into the limitations of the proposed method, such as the cases where the SelfHDR method may not provide optimal results. An exploration of the generalizability limitations of the method would have been appreciated.

**Questions:**

- How sensitive is the SelfHDR method to the quality of the input images (for example, noise)? Would substantial noise or minor shifts in alignment between images drastically affect the performance of the method? This might be a bit hard to comment on, as the misalignments in the Kalantari dataset are already fixed.
- Have the authors considered combining the SelfHDR method with other techniques (perhaps pre-processing or post-processing techniques) to enhance its performance? For example, use a pre-trained denoising network to process the low-exposure image and use a hallucination network such as Stable Diffusion to generate the content in the over-exposed regions.
- Can the authors provide more insights into the cases where the SelfHDR method might fail or provide subpar results? For instance, are there certain types of scenes, color distributions, or specific types of exposure variations that may destabilize the method?

---

> ### Author Response · Authors · 2023-11-22
> **Response to Reviewer TwnM**
>
> We thank the reviewer for the valuable comments and suggestions.
> We appreciate the reviewer's questions and hope our responses could address the concerns.
>
> 1. SelfHDR Sensitivity to Noise
>
> SelfHDR does not specifically deal with the noise problem. If the noise in the multi-exposure image is obvious, the result may also contain noise. Moreover, most existing multi-exposure HDR reconstruction methods (including supervised and self-supervised ones) only focus on removing ghosting artifacts caused by misalignment between inputs, having this limitation as well. SelfHDR has taken a step toward more realistic self-supervised HDR imaging by deghosting, while our ongoing work is to further address this limitation.
>
> Nonetheless, perhaps benefiting from the combination of two complementary supervision, our method still performs better than other self-supervised methods (e.g., FSHDR) when testing noisy images, as shown in Fig. 4(b).
>
>
> 2. SelfHDR Sensitivity to Misalignment
>
>
> The robustness to misalignment depends on the HDR reconstruction networks we use. In Table 1, we use four HDR reconstruction networks (including CNN-based ones and Transformer-based ones) for experiments. From the results, the Transformer-based networks (i.e., HDR-Transformer and SCTNet) perform better. And the qualitative results testing on out-of-domain datasets (i.e., Sen et al. and Tursun et al. datasets) also show this trend.
>
> In addition, benefiting from the self-supervised properties of SelfHDR, the best way to improve robustness may be to collect as much data as possible for training.
>
>
>
> 3. Combining with Other Techniques
>
> For noisy images, we have tried to deploy a pre-trained real-world denoising network, but it leads to over-smooth results, losing many details. This may be because the noise model and level of noisy images mismatch with the data (i.e., SIDD dataset) on which the denoising network was trained.
>
> We also agree with the reviewer's insight that HDR reconstruction can be combined with some image restoration methods, especially when considering more realistic shooting conditions. Our ongoing work is aimed at this. Instead of using pre-trained networks, we think that it may be more appropriate to introduce some self-supervised image restoration works to process input images.
>
> For over-exposed regions, we argue that it is difficult to generate realistic details with a hallucination network such as Stable Diffusion, as it can only imagine the information. Actually, the content of over-exposed regions can be preserved well in low-exposure images. It may be more reliable to search for the corresponding information in low-exposure images.
>
>
> 4. Failed Cases
>
> In addition to handling noisy images, here we give some other situations where SelfHDR may fail. First, when long-exposure images suffer from blur, SelfHDR may introduce blurry results. In future work, we can combine some self-supervised image restoration methods to process the blurry images. Second, when the scene irradiance changes drastically in shooting multi-exposure images, SelfHDR may fail, as the constructed color components may be inaccurate. And this situation is also difficult to handle for supervised methods.
>
>
> Besides, SelfHDR has no specific requirements for scene type and content color. In terms of generalization ability in different scenes, SelfHDR is similar to the corresponding supervised HDR reconstruction networks. And the best way to improve generalization ability is to collect more data and retrain the network.
>
>
> We will add this and give some examples in Sec. E of the appendix in the revision.

---

> > ### Comment · Reviewer_TwnM · 2023-11-22
> > **Thank you!**
> >
> > Thank you so much for addressing my concerns. I have no further questions and would like to stick to my initial positive opinion.

---

### Official Review · Reviewer_nVjt · 2023-10-28

**Soundness:** 4 excellent
**Presentation:** 3 good
**Contribution:** 4 excellent
**Rating:** 8
**Confidence:** 5

**Summary:**

This work proposes a self-supervised HDR reconstruction method SelfHDR for dynamic scenes, which only requires input multi-exposure images during training. Specifically, SelfHDR decomposes latent ground-truth into constructible color and structure component supervisions. Experiments show that SelfHDR achieves comparable performance to supervised ones.

**Strengths:**

1.This work proposes a self-supervised HDR reconstruction method, addressing the problem of the difficult collection of paired HDR data.

2.The idea of constructing color and structure supervision respectively is somewhat novel.

3.This work achieves competitive results compared to supervised methods.

**Weaknesses:**

1.Is the performance upper limit of this work limited by the upper limit of structure or color supervision? What designs correspond to solving this problem?

2.This method may also be limited by the alignment method.

3.Some visualizations of structure and color supervision can be given.

4. Discussion with some traditional multiple exposure fusion methods, such as HDR plus?

**Questions:**

Please see the above weakness.

---

> ### Author Response · Authors · 2023-11-22
> **Response to Reviewer nVjt**
>
> We thank the reviewer for the valuable comments and suggestions.
> We appreciate the reviewer's questions and hope our responses could address the concerns.
>
> 1. Performance Upper Limit
>
> The performance upper limit of our work is indeed limited by the upper limit of structure or color supervision. Actually, the performance of existing supervised methods is also limited by the quality of ground truth (GT). Moreover, although the individual structure or color component is not sufficient as supervision, combining the two components as supervision achieves comparable results to supervised methods. As a self-supervised method, the performance has been enough good.
>
> Below we describe in detail the combination of the two complementary components. Two strategies are presented to combine their advantages and discard their disadvantages. First, we use different loss terms for the two supervisions to learn their different information. We adopt L1 loss for the color component and VGG loss for the structure component. The purpose of the former is to learn color information, while the latter is to learn texture information. Second, we design masks to exclude ghosting areas in color components, thus avoiding adverse impacts from these areas.
>
>
> 2. Limited by Alignment Method
>
> The alignment approach mainly influences the color component. Nevertheless, the color component is not very demanding for the alignment approach, as its poorly aligned areas can be further supplemented by the structure component. Furthermore, the development of the current image alignment works is mature, they provide us with enough assistance to achieve comparable performance to supervised methods.
>
>
> We will add this in Sec. E of the appendix in the revision.
>
> 3. Visualizations of Structure and Color Supervision
>
> We respectively visualize a color and structure supervision in Fig. A (g) and (l) of the appendix.
>
>
> 4. Discussion with HDR+
>
> Traditional methods to remove ghosting include rejecting misaligned areas, aligning input images, and using patch-based composite, as mentioned in the second paragraph of Sec. 1 (Introduction). Recently, deghosting methods based on deep learning perform more effectively than traditional ones. Thus, in this work, we adopt learning-based methods to explore self-supervised HDR reconstruction.
>
> In particular, HDR+ takes bursts captured by the same exposure setting as input, while our method adopts multi-exposure images. Unlike multi-exposure images, bursts have limited dynamic range, which makes HDR+ only suitable for scenes with moderate dynamic range, rather than high dynamic range. In addition, HDR+ may only handle small misalignment between the input images, while our method can handle large motions.

---

### Official Review · Reviewer_Vk35 · 2023-10-30

**Soundness:** 3 good
**Presentation:** 3 good
**Contribution:** 2 fair
**Rating:** 6
**Confidence:** 4

**Summary:**

This paper proposes a self-supervised method to fuse multi-exposed images in dynamic scenes. It learns a reconstruction network under the supervision of two complementary components, including the color component and the structure component. The color component is estimated from aligned multi-exposure images. The structure one is generated through a structure-focused network that is supervised by the color component and an input image. These components construct a pseudo reference for training.

**Strengths:**

1. It is a self-supervised method, which overcomes the need for supervised labeled data in supervised methods.
2. Experiments show that our SelfHDR outperforms the state-of-the-art self-supervised methods, and achieves comparable performance to supervised ones.
3. The way of decomposing an image into color and structure components is new.

**Weaknesses:**

1. The drawback of existing self-supervised methods is that they construct pseudo-pairs for HDR reconstruction. The performance of these methods is unsatisfactory, as their simulated pairs still have gaps with real-world ones. The proposed self-supervised way is also realized by constructing a pseudo reference. Why does this method show advantages? The theoretical discussion and differentiation with existing self-supervised methods are not very sufficient.
2. For dynamic scenes, the method relies on existing registration methods to obtain more accurate color components for registration.
3. This method constructs a pseudo ground truth for self-supervision. The keys lie in the combination optimization (for color components), and mask-based weighting (for structure components). The construction process is simple and mainly based on the functions in Figure 1 which is set by prior.
4. One of the contributions is the construction of a color component. However, it lacks the comparison of the overall results which can directly reflect this advantage in terms of color.

**Questions:**

1. Does this decomposition way (decomposing an image into color and structure components) have a corresponding theoretical basis, and does it correspond to some existing image decomposition theory?
2. “Regardless of the ghosting areas, the rest can record the rough color value, and in which well-aligned ones can offer both good color and structure cues of HDR images”. The experiment lacks the HDR results in the presence of alignment errors.
3. The mask-based weighting does not consider pixel neighborhood relationships. Will it lead to artifacts?

---

> ### Author Response · Authors · 2023-11-22
> **Response to Reviewer Vk35**
>
> We thank the reviewer for the valuable comments and suggestions.
> We appreciate the reviewer's questions and hope our responses could address the concerns.
>
> 1. Discussion with Existing Self-Supervised Methods
>
> Existing self-supervised methods (e.g., FSHDR (Prabhakar et al., 2021) and Nazarczuk et al. (Nazarczuk et al., 2022)) construct pseudo-inputs and a corresponding pseudo-target. For these methods, motion and illumination in input images are synthetic, and exhibit gaps with real-world ones.
>
> In this work, we maintain the original inputs and only construct pseudo-targets for them. It does not destroy the real-world motion and illumination in the original inputs. Moreover, we construct two complementary supervisions to address the possible information missing problem in a single pseudo-target.
>
> We will make it clear in the revision.
>
> 2. Dependence on Registration Method
>
> The color component indeed relies on the registration method. Nevertheless, it is not very demanding for the registration method, as its poorly aligned areas can be further supplemented by the structure component. Furthermore, the development of the current image registration works is mature, they provide us with enough assistance to achieve comparable performance to supervised methods.
>
>
> We will add this in Sec. E of the appendix in the revision.
>
>
> 3. More Discussion about Color Components
>
> The color component is generated by weighting the aligned images, as shown in Eqn. (5). When some areas are aligned well, the corresponding areas in the color component are consistent with the ground-truth. When some areas are aligned poorly, the corresponding areas in the color component may have ghosting artifacts. For the former situation, we can directly take these well aligned areas as supervision of HDR reconstruction network. For the latter situation, we construct a mask in Eqn. (14) to exclude these poorly aligned areas when training HDR reconstruction network. And experiments in Table 4 show that the mask indeed plays a positive role.
>
>
> In addition, we visualize a color component in Fig. A (g) of the appendix. It can be seen that its characteristics are consistent with the phenomenon we describe above.
>
>
>
> 4. Decomposition Way
>
> Our decomposition way is a focus or emphasis on color and structure relatively, not an absolute separation. Nevertheless, it may have some similarities with Laplace decomposition. The color component is similar to the low-frequency part of  Laplace decomposition, while the structure component is similar to the high-frequency part of Laplace decomposition.
>
> 5. Mask-Based Weighting
>
> Fig. 1 is a commonly used weighting function for generating an HDR image from static multi-exposure images. In Fig. 1, the abscissa of the function figure is the medium-exposure image value, and the ordinate is the mask value. In other words, the mask value depends on the image value. As long as the images are aligned well, additional artifacts will not be introduced in the color component. Moreover, we visualize a mask in Fig. A (i) of the appendix. It can be seen the mask is highly relevant to the image value, and is not scattered and irregular.

---

### Meta-Review · Area_Chair_UFzG · 2023-12-07

**Metareview:**

This paper proposes a self-supervised method for HDR image reconstruction in dynamic scenes. The proposed method involves learning a reconstruction network under the supervision of two complementary components: color and structure. A pseudo target for training is constructed using these components.

The proposed method is self-supervised and removes the requirement for paired ground-truth HDR images, which can be challenging to obtain. A self-supervised approach is more practical and may yield better results by utilizing more data. Regarding technical contributions, the method of decomposing an image into color and structure components is new. Several experiments demonstrate that the proposed method outperforms state-of-the-art self-supervised methods and is comparable to supervised methods.

The idea of using two complementary supervisions to construct pseudo targets provides good results, but it lacks rigorous theoretical support. The reviews also question the limitations of relying on alignment methods. Although the rebuttal claims that this is not very demanding for the registration method due to the fact that poorly aligned areas can be supplemented with the structure component, it would be more convincing if some experiments were provided to support this claim. A broader experimental validation would be beneficial to the paper. This paper essentially uses the same datasets as the supervised method. Because the method is self-supervised, it has the potential to produce better results by exploring a large number of multi-exposed images without the reference HDR images. Unfortunately, the potential of this approach is not fully explored by training on more data in the paper. In addition, the paper could be strengthened by testing on more challenging real-world scenarios, especially dealing with complex situations and variable lighting conditions, as one reviewer pointed out.

**Justification For Why Not Higher Score:**

In the rebuttal, it was mentioned that the best way to improve generalization ability is to collect more data and retrain the network. Unfortunately, the paper does not fully validate the potential of the self-supervised method by training on a wider set of data. It would also be helpful if the paper could provide experiments to validate the possible limitations, such as the dependence on alignment.

**Justification For Why Not Lower Score:**

The proposed self-supervised method is a promising alternative for HDR image reconstruction as it demonstrates competitiveness with supervised methods while significantly reducing data requirements.

---

### Decision · Program_Chairs · 2024-01-16

Accept (poster)